# Crosstalk between repair pathways elicits double-strand breaks in alkylated DNA and implications for the action of temozolomide

Robert P Fuchs[1][†]*, Asako Isogawa[2], Joao A Paulo[3], Kazumitsu Onizuka[4], Tatsuro Takahashi[5], Ravindra Amunugama[1], Julien P Duxin[1][‡], Shingo Fujii[2]

[1]Department of Biological Chemistry and Molecular Pharmacology, Harvard Medical School, Boston, United States; [2]Cancer Research Center of Marseille, UMR7258, CNRS, Marseille, France; [3]Department of Cell Biology, Harvard Medical School, Boston, United States; [4]Institute of Multidisciplinary Research for Advanced Materials, Tohoku University, Sendai, Japan; [5]Faculty of Science, Kyushu University, Fukuoka, Japan

*For correspondence:
robert.fuchs@inserm.fr

Present address: [†]Marseille Medical Genetics, UMR1251, Marseille, France; [‡]The Novo Nordisk Foundation Center for Protein Research, University of Copenhagen, Copenhagen, Denmark

**Competing interests:** The authors declare that no competing interests exist.

**Abstract** Temozolomide (TMZ), a DNA methylating agent, is the primary chemotherapeutic drug used in glioblastoma treatment. TMZ induces mostly N-alkylation adducts (N7-methylguanine and N3-methyladenine) and some $O^6$-methylguanine ($O^6$mG) adducts. Current models propose that during DNA replication, thymine is incorporated across from $O^6$mG, promoting a futile cycle of mismatch repair (MMR) that leads to DNA double-strand breaks (DSBs). To revisit the mechanism of $O^6$mG processing, we reacted plasmid DNA with N-methyl-N-nitrosourea (MNU), a temozolomide mimic, and incubated it in *Xenopus* egg-derived extracts. We have shown that in this system, MMR proteins are enriched on MNU-treated DNA and we observed robust, MMR-dependent, repair synthesis. Our evidence also suggests that MMR, initiated at $O^6$mG:C sites, is strongly stimulated in cis by repair processing of other lesions, such as N-alkylation adducts. Importantly, MNU-treated plasmids display DSBs in extracts, the frequency of which increases linearly with the square of alkylation dose. We suggest that DSBs result from two independent repair processes, one involving MMR at $O^6$mG:C sites and the other involving base excision repair acting at a nearby N-alkylation adduct. We propose a new, replication-independent mechanism of action of TMZ, which operates in addition to the well-studied cell cycle-dependent mode of action.

## Introduction

Alkylating agents, a class of important environmental carcinogens, have been widely used in molecular biology to study fundamental repair processes and in the clinic to treat cancer patients. Among the DNA adducts produced by methylating agents such as N-methyl-N-nitrosourea (MNU) and temozolomide (TMZ), a clinically used mimic, the most abundant are two N-alkylation adducts, at the N7 position of guanine (7mG: 70–75% of total alkyl adducts) and the N3 position of adenine (3mA: 8–12%). Importantly, both reagents also produce 8–9% O-alkylation adducts in the form of $O^6$-methylguanine ($O^6$mG). This feature contrasts with another common methylating agent, methyl-methane sulfonate (MMS), which forms a much lower level of $O^6$mG (<0.3%) while producing similarly high proportions of 7mG (81–83%) and 3mA (10–11%) (*Beranek, 1990*). For many years, the differences in O versus N reactivities have been rationalized by differences in chemical reaction mechanisms; on one side, compounds such as MMS, with very low O-reactivity, were classified as SN2 agents (bimolecular nucleophilic substitution) while other agents, such as MNU and TMZ, with

increased O adduct formation, were called SN1 agents (monomolecular nucleophilic substitution). While this classification turned out not to be mechanistically accurate (*Loechler, 1994*), we will nevertheless use this nomenclature throughout this paper for the sake of simplicity. The major N-alkylation (N-alkyl) adducts (7mG and 3mA) are repaired by base excision repair (BER), using N-methylpurine DNA glycosylase (MPG), also known as 3-alkyladenine DNA glycosylase (AAG), and alkylpurine DNA N-glycosylase (APNG) (*Chakravarti et al., 1991*; *Lindahl, 1976*). O-alkylation adducts ($O^6mG$, $O^4mT$) can be directly repaired by $O^6$-methylguanine DNA methyl transferase (MGMT), a protein that transfers the methyl group from these adducts to one of its cysteine residues (*Demple et al., 1982*; *Olsson and Lindahl, 1980*; *Tano et al., 1990*). In addition, alkylating agents also produce a variety of other minor (1–2%) N-alkyl adducts, namely 1mA, 3mC, 3mT, and 1mG that are directly demethylated by AlkB homologs (*Aas et al., 2003*; *Duncan et al., 2002*; *Falnes et al., 2002*; *Trewick et al., 2002*). In summary, SN1 and SN2 alkylating agents produce a diverse array of DNA adducts, but they differ greatly in the amount of $O^6mG$ produced.

Agents such as MMS mostly induce N-alkyl adducts that lead to DSBs during S-phase as a consequence of BER repair. Indeed, inactivation of the AAG glycosylase, the BER-initiating enzyme, suppresses DSB while inactivation of Polβ leads to their exacerbation (*Simonelli et al., 2017*; *Tang et al., 2011*; *Trivedi et al., 2005*). In rodent cells, it was proposed that MMS-induced DSBs arise when replication meets BER-induced single-strand breaks (SSBs) (*Ensminger et al., 2014*). The toxicity of N-alkyl adducts was found to depend on cell type. AAG-mediated repair of N-alkyl adducts was found to mitigate toxicity in mouse ES cells and HeLa cells, while repair was shown to cause toxic intermediates in retina and bone marrow cells (*Meira et al., 2009*). In all cell types, O-alkyl adducts were found to be highly cytotoxic and mutagenic. While the mutagenicity of $O^6mG$ is easily accounted for by its high propensity to mispair with T during DNA synthesis (*Bhanot and Ray, 1986*; *Loechler et al., 1984*; *Mazon et al., 2010*), its cytotoxicity is intriguing since $O^6mG$ per se does not interfere with DNA synthesis. A seminal paper, published 50 years ago by *Plant and Roberts, 1971*, noted that when synchronized HeLa cells are treated in G1 with MNU, they continue through the first cell cycle almost normally and with little effect on DNA synthesis. On the other hand, there is a dramatic effect on DNA synthesis in the second cell cycle after MNU exposure. These data led the authors to surmise that cytotoxicity stems from a secondary lesion that forms when DNA synthesis occurs across $O^6mG$ template adducts (*Plant and Roberts, 1971*). It was demonstrated later that MNU-mediated inhibition of DNA synthesis, in the first and the second cycle, is due to the action of the MMR machinery that acts on $O^6mG$:T lesions that form upon DNA synthesis (*Goldmacher et al., 1986*; *Kat et al., 1993*; *Noonan et al., 2012*; *Plant and Roberts, 1971*; *Quiros et al., 2010*).

Indeed, $O^6mG$:T lesions were found to be excellent substrates for MMR (*Duckett et al., 1999*; *Yoshioka et al., 2006*). During MMR gap-filling, the $O^6mG$:T mispair is reformed, potentially leading to another round of MMR, thus entering so-called futile MMR cycles (*Kaina et al., 2007*; *Karran and Bignami, 1994*; *Olivera Harris et al., 2015*; *York and Modrich, 2006*). The MMR cycling model has received experimental support in vitro (*York and Modrich, 2006*) and in *Escherichia coli* (*Mazon et al., 2010*). Studies with synchronized cells have shown that the critical events related to cytotoxicity occur in the second cell cycle post-treatment (*Quiros et al., 2010*). However, as discussed in recent review articles, the precise mechanism by which MMR leads to DSBs has yet to be established (*Gupta and Heinen, 2019*; *Kaina and Christmann, 2019*).

While most studies have been devoted to MNU-induced cell cycle effects, in the present paper we wanted to investigate the early response to MNU treatment, that is, in the absence of replication. We addressed this question using *Xenopus* egg-derived extracts, which recapitulate most forms of DNA repair (*Wühr et al., 2014*). Upon incubation in these extracts, plasmids treated with MNU exhibit robust repair synthesis in the absence of replication. Repair synthesis occurs at $O^6mG$:C lesions, depends on MMR, and involves an excision tract of several hundred nucleotides. MMR events at $O^6mG$:C sites are robustly stimulated by additional processing at N-alkylation lesions, most likely via BER. Previous studies have described activation of MMR in the absence of replication in cells treated by SN1-methylating agents, a process termed noncanonical MMR (ncMMR) (*Peña-Diaz et al., 2012*). Interestingly, we observed replication-independent induction of DSBs in MNU-treated plasmids. The kinetics of DSB formation obeys a quadratic MNU dose-response, suggesting the involvement of two independent repair events. We propose that DSBs occur when the gap

generated at an $O^6$mG adduct during MMR overlaps with a BER intermediate initiated at an N-alkyl adduct in the opposite strand.

These data reveal a novel facet of MNU-induced damage to DNA that is replication independent. Extrapolation of the in vitro data led us estimate that $\approx$ 10 DSBs per cell can be induced by a single daily dose of TMZ used in the clinic in the absence of replication.

## Results

### Reaction conditions leading to similar levels of DNA alkylation

Our goal is to determine the DNA proteome for distinct alkylating agents. For the sake of comparison, we needed to determine the reaction conditions for different alkylating agents that lead to similar levels of total alkylation. As a proxy for total alkylation, we monitored the amount of N-alkyl adducts, namely 7mG and 3mA, that together represents >80% of alkylation for MMS and MNU. Estimation of the N-alkyl adduct level is achieved by converting these adducts to single-stranded DNA (ssDNA) breaks by a combination of heat depurination and alkali cleavage treatments (*Maxam and Gilbert, 1977*; *Figure 1—figure supplement 1A*). The resulting plasmid fragmentation patterns were resolved and analyzed by agarose gel electrophoresis. The reaction conditions were adjusted (by trial and error) as to generate a median fragment size of 500 nt, corresponding to one alkylated base every 500 nucleotides on average (*Figure 1—figure supplement 1A*).

### Identification of the proteins that specifically bind to DNA alkylation damage in nucleoplasmic extracts

In order to identify the proteins binding to $O^6$mG-containing base pairs in *Xenopus* egg-derived extracts, we used a recently developed plasmid pull-down procedure, IDAP, for the identification of DNA-associated proteins (*Isogawa et al., 2020*; *Isogawa et al., 2018*). As outlined above, MNU produces twenty- to twentyfivefold more $O^6$mG lesions than MMS (0.3% and 7–8% of total alkylation, respectively), while the relative amounts of N-alkyl lesions produced by the two agents are similar (>80% of N7mG+N3mA) (*Beranek, 1990*). These agents react chemically with DNA under neutral pH conditions, and we established in vitro reaction conditions that trigger comparable levels of plasmid alkylation (see above and *Figure 1—figure supplement 1A*).

The pull-down procedure involves immobilization of plasmid DNA on magnetic beads by means of a triple helix-forming probe (*Figure 1A*; *Isogawa et al., 2020*; *Isogawa et al., 2018*). The same amount of untreated or alkylated plasmids was coupled to magnetic beads and incubated in nucleoplasmic extracts (NPE) derived from *Xenopus* eggs (*Walter et al., 1998*). The reaction was stopped by dilution into a formaldehyde-containing buffer, which fixes protein-DNA complexes. After washing the beads and reversing the cross-links, the recovered proteins were visualized by silver staining following sodium dodecyl sulphate–polyacrylamide gel electrophoresis (SDS-PAGE) (*Figure 1—figure supplement 1B*). As a negative control, mock-conjugated beads (noDNA control lane) exhibit a low-protein background, illustrating efficient removal of non-specific proteins (*Figure 1—figure supplement 1B*). Proteins captured on the different plasmid samples were analyzed by label-free mass spectrometry (MS) as described in 'Materials and methods'. The MS data are presented in the form of volcano plots. When comparing MNU-treated to undamaged control plasmids, the MMR proteins (labeled in red) were highly enriched in the MNU sample (*Figure 1B*). All six canonical MMR proteins (MSH2, MSH3, MSH6, MLH1, PMS1, and PMS2) were specifically enriched on MNU plasmids. These proteins form the MutSα, MutSβ, MutLα, and MutLβ heterodimers (*Jiricny, 2006*). Other proteins known to participate in MMR, RAD18, POLη, EXO1, and two subunits of Pol delta (POLD2 and POLD3), were also specifically enriched on MNU plasmids. Previously, it was shown that purified MutSα does not bind to $O^6$mG:C base pairs (*Yoshioka et al., 2006*). Our present experiments involve extracts containing many proteins, and there is probably synergy between MutSα and MutLα (and other proteins) to achieve full MMR (*Ortega et al., 2021*). Activation of MMR by a single $O^6$mG:C lesion has been reported previously (*Duckett et al., 1999*).

It was previously noted that upon oxidative stress, produced by hydrogen peroxide treatment, RAD18 and POLη are recruited to chromatin in a MSH2-MSH6 (MutSα)-dependent manner in human cells (*Zlatanou et al., 2011*). While MutSα, MutSβ, and MutLα functionally participate in MMR, the role of MutLβ (MLH1-PMS1) remains unknown (*Jiricny, 2006*). No MMR proteins were enriched on

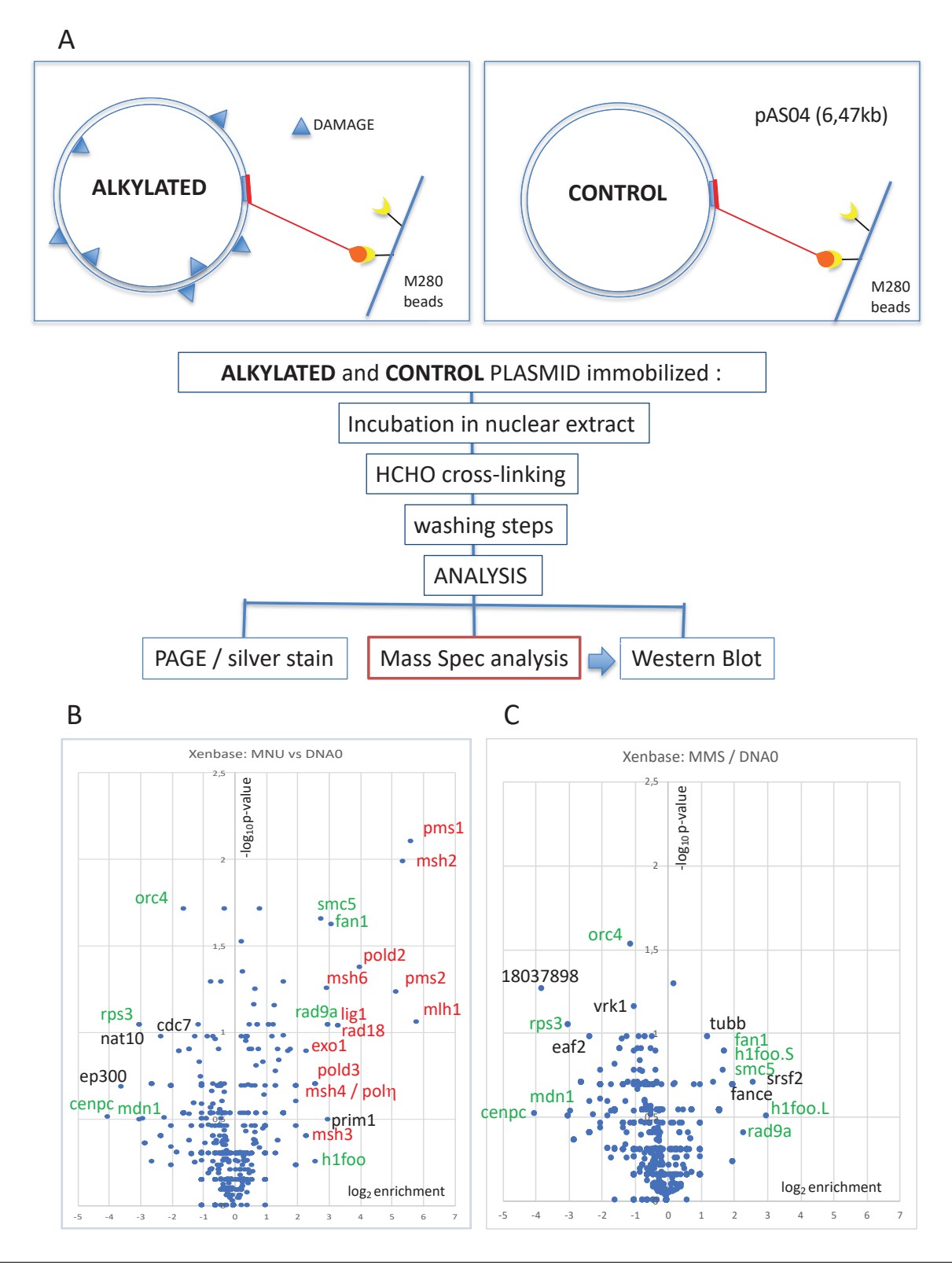

**Figure 1.** Pull-down of proteins that bind to alkylated versus untreated plasmid DNA. (**A**) Experimental workflow. Plasmid DNA (pAS04, 6.5 kb) was treated with alkylating agents under conditions leading to a similar extent of N-alkylation (≈ one alkaline cleavage site every 500 nt) (**Figure 1—figure supplement 1A**). Immobilized plasmid DNA was incubated in *Xenopus* nucleoplasmic extracts (NPE) for 10 min at room temperature under mild agitation. The reaction was stopped by addition of formaldehyde (0.8% final) to cross-link the protein-DNA complexes. The beads were processed and
*Figure 1 continued on next page*

*Figure 1 continued*

analyzed by polyacrylamide gel electrophoresis (PAGE) or by mass spectrometry (MS) as described in 'Materials and methods'. (**B**) Relative abundance of proteins captured on N-methyl-N-nitrosourea (MNU)-treated versus -untreated DNA0. Proteins captured on equal amounts of MNU-treated or -untreated plasmid were analyzed by label-free MS in triplicate. For all proteins, average spectral count values in the MNU-treated plasmid sample were divided by the average spectral count values in the DNA0 sample. The resulting ratio is plotted as its $\log_2$ value along x-axis. The statistical significance of the data is estimated by the p-value in the Student's t-test and plotted as $-\log_{10}p$ along y-axis. Proteins enriched on MNU versus untreated plasmid DNA appear on the right-side top corner and essentially turn out to be mismatch repair (MMR) proteins labeled in red (**B**). Data shown are analyzed using Xenbase database. (**C**) Relative abundance of proteins captured on methyl-methane sulfonate (MMS)-treated versus -untreated DNA0. Proteins captured on equal amounts of MMS-treated or -untreated plasmid were analyzed by label-free MS in triplicate. The data are analyzed and plotted as in panel (**B**) for MNU using Xenbase database. Proteins (labeled in green in **B** and **C**) are found enriched or excluded in both MMS versus DNA0 and MNU versus DNA0 plasmids. We suggest these proteins are recruited or excluded from binding to DNA by the abundant class of N-alkylation adducts that both MMS- and MNU-treated plasmids share in common (~27 N-alkyl adducts per plasmid).

The online version of this article includes the following figure supplement(s) for figure 1:

**Figure supplement 1.** Alkylation reaction conditions and differential protein capture data.

MMS-treated plasmids (*Figure 1C*). As MNU treatment induces 20–30 times more $O^6mG$ adducts than MMS, we postulate that recruitment of MMR proteins depends on $O^6mG$. Comparison of proteins captured on MNU- versus MMS-treated plasmids indeed reveals specific enrichment of MMR proteins. Proteins specifically recruited at N-alkyl adducts (in green in *Figure 1B and C*) are absent in the MMS versus MNU volcano plot (*Figure 1—figure supplement 1C*), since N-alkyl adducts are equally present in both MMS- and MNU-treated plasmids.

In addition, compared to the lesion-free control plasmid, some proteins were enriched on or excluded from both MMS- and MNU-treated plasmids (*Figure 1B and C*, green labels). We suggest that the recruitment or exclusion of these proteins depends on the abundant 7mG and 3mA adducts formed by both MMS and MNU. The reason why BER proteins, normally involved in the repair of these N-alkyl adducts, were not captured is unclear. One possibility is that BER proteins interact too transiently with DNA to be efficiently captured.

## Repair of alkylated plasmid DNA in NPE

We next investigated the repair of DNA treated by the different alkylating agents in NPE. Plasmids were alkylated with MMS, MNU, or ENU to a density of one lesion every ≈500 nt (*Figure 1—figure supplement 1A*). The alkylated plasmids were incubated in NPE in the presence of $\alpha^{32}$P-dATP. These extracts contain high levels of geminin, an inhibitor of replication licensing. Therefore, any observed DNA synthesis occurs independently of DNA replication and corresponds to the so-called 'unscheduled DNA synthesis' (UDS) (*Figure 2A*). Undamaged plasmids exhibited a low level of background DNA synthesis, whereas MNU- and ENU-treated plasmids sustained robust, time-dependent UDS equivalent to 3–4% of the synthesis needed for a full round of replication (*Figure 2B*). MMS-treated plasmids exhibited UDS that was just twofold above the background seen in undamaged plasmids (*Figure 2B*). Given that the assay measures incorporation of $\alpha^{32}$P-dATP, long-patch BER events (*Sattler et al., 2003*) will be detected, while short-patch BER events (1 nt patch) will only be detected at 3mA but not at 7mG adducts. The assay is clearly biased toward the detection of events such as MMR that involve repair patches hundreds of nucleotides long.

We asked whether the observed UDS in MNU- and ENU-treated plasmids was MMR dependent, as suggested by the MS results. To test this idea, we depleted MMR proteins from extracts using antibodies (*Figure 2—figure supplement 1A*), whose specificity was previously validated (*Kato et al., 2017*; *Kawasoe et al., 2016*). Depletion of MLH1 or PMS2 severely reduced UDS in MNU-treated plasmids, while no reduction was observed in PMS1-depleted extracts (*Figure 2—figure supplement 1B*). This observation is consistent with the fact that MutLα (composed of MLH1 and PMS2) is involved in canonical mismatch repair whereas MutLβ (composed of MLH1 and PMS1) is not (*Jiricny, 2006*). Aphidicolin (Aph), an inhibitor of B-family DNA polymerases (*Baranovskiy et al., 2014*), decreased incorporation, on average, by 3.5-fold on MNU and ENU plasmids while it had a more modest effect on MMS-treated plasmids (1.5-fold) (*Figure 2—figure supplement 1C*). These results support the notion that UDS on MNU- and ENU-treated plasmids involves MMR, including a gap-filling event that most likely depends on DNA Polδ, the only B family polymerase detected in the MS analysis described above. Short-patch BER events are mediated by

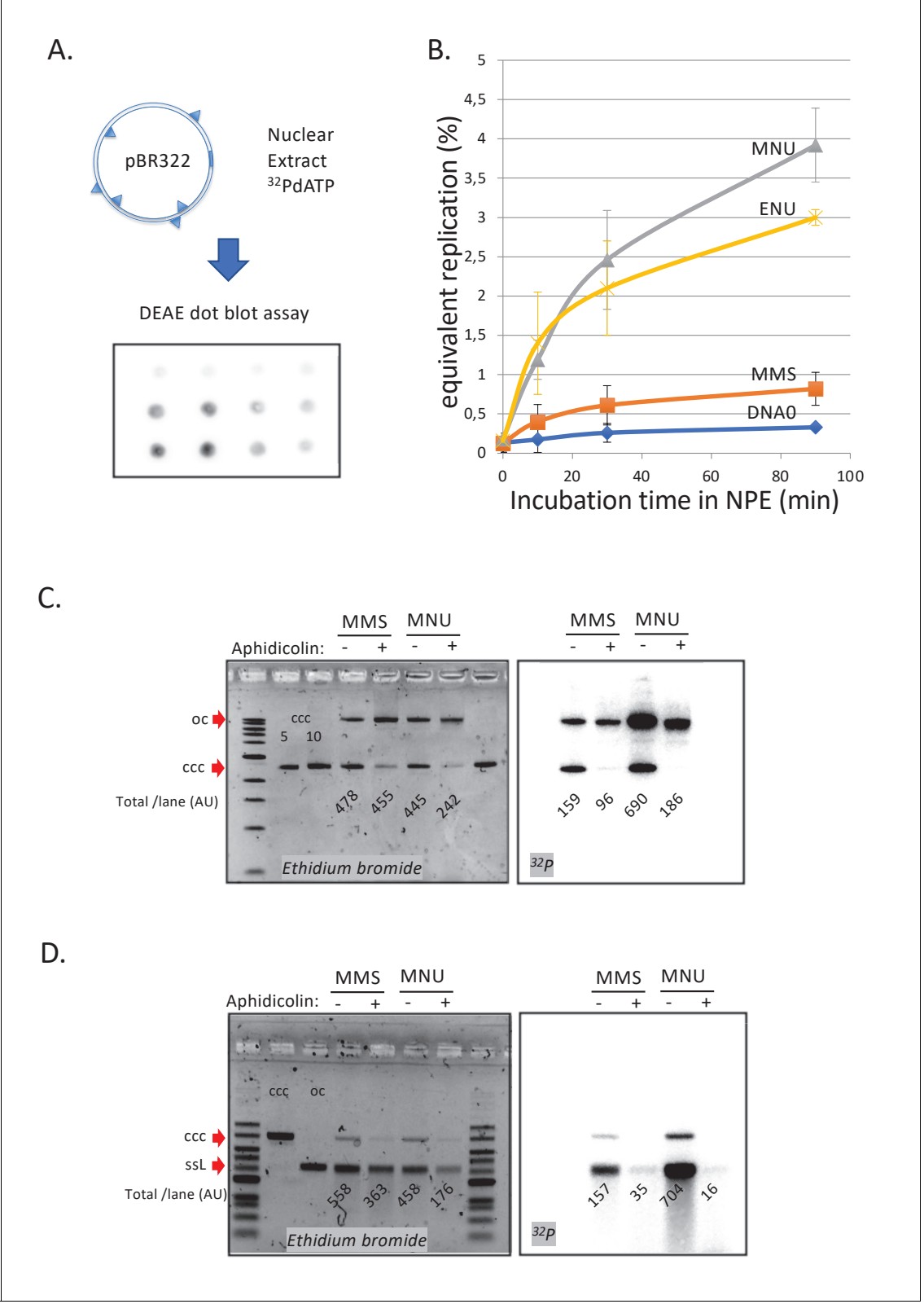

**Figure 2.** DNA repair synthesis in alkylated and undamaged control plasmid DNA in NPE. (**A**) Outline of the spot assay. Plasmids were incubated in nuclear extracts supplemented with α³²P-dATP; at various time points, an aliquot of the reaction mixture was spotted on DEAE paper (see 'Materials and methods'). The dot blot is shown for the sake of illustration only. (**B**) Plasmid DNA pBR322 (4.3 kb) samples, modified to a similar extent with -MMS, -MNU and -ENU, were incubated in nucleoplasmic extracts (NPE) supplemented with α³²P-dATP at room temperature; incorporation of radioactivity

*Figure 2 continued on next page*

*Figure 2 continued*

was monitored as a function of time using the spot assay described above (**A**). Undamaged plasmid DNA0 was used as a control. At each time point, the average values and standard deviation from three independent experiments were plotted. The y-axis represents DNA repair synthesis expressed as a fraction of input plasmid replication (i.e., 10% means that the observed extent of repair synthesis is equivalent to 10% of input plasmid replication). This value was determined knowing the average concentration of dATP in the extract (50 µM) and the amount of added $\alpha^{32}$P-dATP. (**C**) N-methyl-N-nitrosourea (MMS)- and N-methyl-N-nitrosourea (MNU)-treated plasmids were incubated in NPE, supplemented or not, by aphidicolin (150 µM final). After 1 hr of incubation, plasmids were purified and analyzed by agarose gel electrophoresis under neutral loading conditions. The gel was imaged by fluorescence (left: ethidium bromide image) and by autoradiography (right: $^{32}$P image). The number below each lane indicates the total amount of signals per lane (expressed in arbitrary units [AU]). Aphidicolin treatment decreases incorporation into MNU-treated plasmid close to fourfold, while it affected incorporation into MMS-treated plasmid only 1.6-fold. (**D**) Samples as in (**C**). Gel loading is performed under alkaline conditions to denature DNA before entering the neutral agarose gel, allowing single-stranded nicks present in DNA to be revealed. The number below each lane indicates the amount of signals per lane (AU).

The online version of this article includes the following figure supplement(s) for figure 2:

**Figure supplement 1.** Involvement of mismatch repair in repair synthesis and effect of aphidicolin.

**Figure supplement 2.** Repair synthesis in HSS extracts.

Polβ (X family), which are insensitive to aphidicolin. The modest sensitivity of MMS-induced UDS to aphidicolin is probably due to a fraction of BER events that belong to the long-patch BER pathway mediated by Polδ/ε (*Sattler et al., 2003*).

We wanted to estimate the average amounts of DNA synthesis associated with MMR at $O^6$mG:C sites and BER at N-alkyl sites, respectively. At the 90 min time point (i.e., at near-plateau value), the difference in UDS between MNU- and MMS-treated plasmids, that is attributable to repair at $O^6$mG:C sites, was equivalent to $\approx 3.1\%$ of the input DNA (*Figure 2B*) or $\approx 270$ nt (pBR322 plasmid is 4363 bp long). With an estimated $\approx 1.7$ $O^6$mG adducts per plasmid, the average repair patch per $O^6$mG adduct is $\approx 160$ nt provided all $O^6$mG lesions are targeted by MMR. Evidence obtained with G:T and $O^6$mG:T constructs (see below) indicates that, under present experimental conditions, only about $\approx 30\%$ of $O^6$mG are substrates for MMR, suggesting that, on average, an MMR patch is $\approx 500$ nt long. Importantly, the MGMT inhibitor Patrin-2 had no effect on UDS of MNU-treated plasmid, even at a dose of 200 µM (data not shown). Surprisingly, inhibition of MGMT by Patrin-2 was previously shown to occur in *Xenopus* extracts (*Olivera Harris et al., 2015*). Two possibilities may account for the lack of any measurable effect of MGMT inhibition: (i) the number of MGMT molecules present in the extract is small compared to the number of $O^6$mG lesions introduced in the incubation mix or (ii) our batch of Patrin inhibitor is inactive. In all cases, if partial demethylation of $O^6$mG by MGMT occurs, the observed amount of UDS would be under-estimated. Thus, the conclusion reached in the paper, namely that $O^6$mG:C sites are substrates for MMR, remains correct.

With respect to N-alkyl adduct repair in MMS plasmid, repair synthesis above the lesion-free DNA control is equivalent to $\approx 0.5\%$ of input DNA (*Figure 2B*), corresponding to 43 nt total synthesis per plasmid. With $\approx 17$ N-alkyl adducts per plasmid, the average DNA synthesis patch per adduct, in case all N-alkyl lesions are repaired, is $\approx 2.6$ nt, a value consistent with a mixture of long ($\approx 2–8$ nt)- and short-patch (1 nt) BER events at N-alkyl adducts. In summary, the average DNA repair patch sizes at $O^6$mG:C ($\approx 500$ nt) and N-alkyl (2–3 nt) sites are compatible with MMR and BER, respectively.

To learn more about UDS in this system, we analyzed repair products via gel electrophoresis. Plasmid pBR322 treated with MMS or MNU was incubated in NPE, supplemented or not with aphidicolin in the presence of $\alpha^{32}$P-dATP, and analyzed on a neutral agarose gel. As already noticed above (*Figure 2—figure supplement 1C*), addition of aphidicolin led to more severe reduction in incorporation into MNU ($\approx 3.7$-fold)- compared to MMS-treated plasmids ($\approx 1.6$-fold) (*Figure 2C*). We also note that in MNU-treated plasmids, in the absence of Aph, open circular repair products were threefold more abundant than closed circular products (*Figure 2C*; $^{32}$P image). This observation suggests that MMR repair was complete in only $\approx 25\%$ of plasmid molecules while 75% of molecules contained at least one nick (or a gap). Interestingly, there was a $\approx 50\%$ loss of total DNA in the MNU+-Aph lane compared to the other lanes, suggesting massive DNA degradation in NPE due to polymerase inhibition by Aph. Indeed, the observed DNA degradation can specifically be linked to repair events as the loss in radioactivity in MNU lanes -Aph versus +Aph is >70% (*Figure 2C*; $^{32}$P image). Under alkaline loading conditions (*Figure 2D*), repair products ($^{32}$P image) in MNU-treated

plasmids appeared mostly as a single-stranded linear band form. In addition, there was a large smear (>25% of material) of shorter fragments. These results show that most plasmids contain one nick and some contain several nicks. In the +Aph samples, the open circular (oc) form, seen in the gel loaded under neutral conditions (*Figure 2C*), disappears under alkaline loading conditions (*Figure 2D*). This suggests that these oc molecules (*Figure 2C*) contain many nicks that run as short fragments upon denaturation. In conclusion, MNU-treated plasmids undergo robust repair synthesis that is more sensitive to aphidicolin inhibition than MMS-treated plasmids.

We next examined $O^6$mG-induced DNA synthesis in a different extract, namely high-speed supernatant (HSS) of total egg lysate. Unmodified pBR322 plasmids (DNA0) or those treated with MNU to an extent of ≈1 N-alkyl adduct/500nt were incubated in the presence of $\alpha^{32}$P-dATP. Repair synthesis was monitored at room temperature (RT) as a function of time using the spot assay described above (*Figure 2A*). In HSS extract, MNU-treated plasmids did not exhibit significant repair synthesis (*Figure 2—figure supplement 2*), in contrast to the robust repair synthesis seen in NPE (*Figure 2B*). Although, HSS contains lower concentrations of most DNA repair enzymes compared to NPE, HSS was shown to be proficient for MMR at a single $O^6$mG provided a nick is present in proximity (*Olivera Harris et al., 2015*). We reasoned that HSS might not contain adequate concentrations of the DNA glycosylase AAG, which initiates BER at N-alkyl sites. When HSS extract was supplemented with purified AAG glycosylase (150 nM) (NEB, Biolabs), robust repair synthesis is observed in MNU-treated plasmids (*Figure 2—figure supplement 2*). These observations suggest the involvement of BER in stimulating MMR at $O^6$mG lesions.

## MMR at single $O^6$mG-containing base pairs is enhanced by the presence of N-alkylation adducts

Next, we explored a possible crosstalk between repair pathways acting on alkylated DNA. In MNU-treated plasmids, there is on average one $O^6$mG adduct for every 9–10 N-alkyl adducts (*Beranek, 1990*). To investigate the repair response triggered by a single $O^6$mG:C lesion alone or in the presence of additional N-alkyl adducts, we implemented a reconstitution experiment. For that purpose, a single $O^6$mG:C construct (mGC) (*Isogawa et al., 2020*) was treated with MMS to introduce ≈9–10 N-alkyl adducts per plasmid molecule, generating plasmid mGC+MMS, which is expected to recapitulate adduct distribution found in MNU-treated plasmids. Control plasmid GC was treated with the same concentration of MMS, to generate GC+MMS. These in vitro manipulations did not affect plasmid topology as all four constructs exhibit a similar migration pattern (*Figure 3—figure supplement 1A*).

Plasmid constructs GC and mGC and the corresponding two MMS-treated constructs (GC+MMS and mGC+MMS) (*Figure 3A*) were incubated with NPE in the presence of $\alpha^{32}$P-dATP to monitor repair synthesis (i.e., UDS). We observed incorporation of radioactivity specifically attributable to the single $O^6$mG:C lesion (compare mGC with GC in *Figure 3B*). Activation of MMR by a single $O^6$mG:C lesion has been reported previously (*Duckett et al., 1999*). The specific involvement of MMR in $O^6$mG-dependent incorporation was re-assessed, by incubating the single adducted $O^6$mG:C construct in MLH1-depleted NPE; radioactive incorporation above background was fully abolished in mGC plasmids (*Figure 3—figure supplement 1E*). How MMR may get engaged in a repair reaction on a closed circular template will be considered in the 'Discussion' section.

Importantly, repair synthesis, due to the single $O^6$mG:C lesion, is strongly enhanced by the presence of MMS adducts (compare mGC+MMS with GC+MMS in *Figure 3B*). At the 2 hr time point, incorporation, above background, due to the single $O^6$mG, expressed in % replication equivalent, represents 0.64% (difference between mGC and GC), while it amounts to 1.85% in the presence of MMS lesions (compare mGC+MMS with GC+MMS). One can thus estimate that, incorporation due to a single $O^6$mG lesion, is stimulated about 2.9-fold (1.85/0.64) by the presence in cis of MMS adducts (*Figure 3B*).

Finally, we wanted to compare the relative MMR repair efficiencies triggered by $O^6$mG:C and $O^6$mG:T (or GT) mismatches (*Figure 3A and B*). These constructs were used as single adducted constructs or, after additional reaction with MMS, similarly to the procedure described for the corresponding GC or mGC constructs. The main observation is that GT-containing constructs trigger a much stronger MMR response than their GC counterparts (*Figure 3B*). In the absence of MMS, at the 120′ time point, the level of UDS in mGC represents 23% of the level in mGT. In the presence of MMS, at the 120′ time point, the level of UDS in mGC+MMS represents 38% of the level in mGT

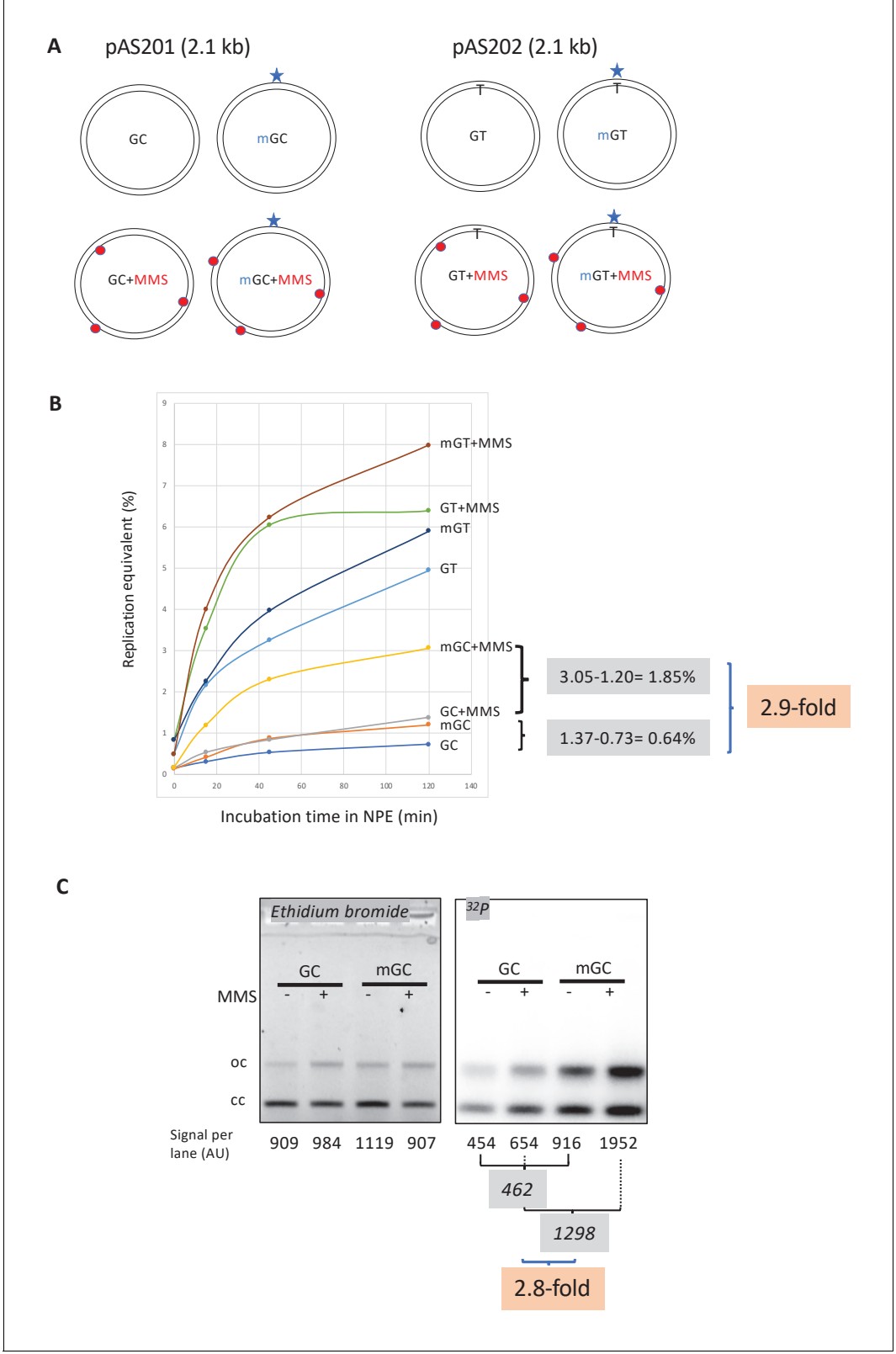

**Figure 3.** Stimulation of MMR at a single O⁶mG site by N-alkyl adducts in cis. (A) Covalently closed circular (ccc) plasmids (pAS200.2, 2.1 kb) containing a site-specific O⁶mG:C base pair (plasmid mGC) and the corresponding lesion-free control (plasmid GC) were constructed (*Isogawa et al., 2020*). Similarly, plasmids with a site-specific GT or a O⁶mG:T mismatch were constructed. All the four constructs were treated with methyl-methane sulfonate (MMS) in order to introduce random N-alkyl (7mG and 3mA) adducts, generating plasmids GC+MMS, mGC+MMS, GT+MMS, and mGT+MMS. We

*Figure 3 continued on next page*

*Figure 3 continued*

adjusted the MMS reaction conditions as to introduce ≈ nine adducts per plasmid (i.e., one N-alkylation adduct every ≈ 500 nt). The resulting proportion of O-alk and N-alkyl adducts mimics the proportion in N-methyl-N-nitrosourea (MNU)-treated plasmids. The single $O^6$mG adduct and the randomly located N-alkyl adducts are represented by a star and red dots, respectively. (**B**) Plasmids described above were incubated in nucleoplasmic extracts (NPE) supplemented with $\alpha^{32}$P-dATP at room temperature; incorporation of radioactivity was monitored as a function of time using the spot assay. The y-axis represents the percentage of DNA repair synthesis with respect to input DNA (i.e., 10% means that the observed extent of repair synthesis is equivalent to 10% of input plasmid replication). Overall, incorporation into GT and mGT plasmids is higher than incorporation into their GC and mGC counterparts. Incorporation attributable to repair at the $O^6$mG:C lesion is increased close to threefold due to the presence of random N-alkyl lesions introduced by MMS treatment. The stimulatory effect of random N-alkyl lesions on GT and mGT repair is observed but is slightly less pronounced than for mGC. (**C**) The same plasmids were incubated for 2 hr in NPE, purified, resolved by agarose gel electrophoresis, and revealed by ethidium bromide fluorescence and $^{32}$P autoradiography. The total amount of signals per lane is indicated (arbitrary units [AU]). As expected, the amount of plasmid extracted from each incubation mix is relatively constant, as quantified below the ethidium bromide image. Increase in repair at the $O^6$mG:C lesion due to MMS treatment (2.8-fold) is in good agreement with data in (**B**).

The online version of this article includes the following figure supplement(s) for figure 3:

**Figure supplement 1.** Mapping repair synthesis in the vicinity of a single $O^6$mG adduct.

---

+MMS. In conclusion, supposing that 100% of mGT mispairs are fully repaired, the extent of mGC repair would be in the range of 30%.

## Nucleotide incorporation occurs in the vicinity of the single $O^6$mG adduct

The plasmids described above were incubated in $\alpha^{32}$P-dATP-supplemented NPE for 2 hr, purified, and analyzed by agarose gel electrophoresis (*Figure 3C*). Covalently closed circular (ccc) and relaxed forms (oc) were quantified in each lane (*Figure 3C*). In the presence of MMS adducts, the single $O^6$mG:C lesion contributes to a 2.8-fold increase in radioactive incorporation compared to its contribution in the absence of MMS (*Figure 3C*) in good agreement with the UDS data (*Figure 3B*).

We wanted to map the repair patches with respect to the $O^6$mG adduct position by restriction enzyme analysis. Digestion of the purified plasmids with *BmtI* and *BaeGI* restriction enzymes generates fragment S (589 bp) that encompasses the $O^6$mG:C site and fragment L (1525 bp) (*Figure 3— figure supplement 1B*). Following separation by agarose gel electrophoresis, the DNA was imaged by ethidium bromide fluorescence and $^{32}$P autoradiography (*Figure 3—figure supplement 1C*). For each fragment, we determined its specific activity by dividing the radioactivity signal by its amount as determined from the ethidium bromide image (*Figure 3—figure supplement 1D*). As expected, the specific activities of S and L fragments were similar in GC (random background incorporation: 0.125±0.015 AU [arbitrary units]) and MMS-treated (GC+MMS) (0.235±0.025 AU) control plasmids. In GC+MMS, the specific activity was slightly higher than in control plasmids, probably reflecting BER-mediated incorporation at randomly distributed N-alkyl adducts. In the two $O^6$mG:C-containing plasmids (mGC and mGC+MMS), the S fragment exhibits a significantly higher specific activity than the L fragment, indicating that MMR activity is centered around the $O^6$mG:C site. In the absence of MMS, incorporation in mGC above background (dotted line in *Figure 3—figure supplement 1D*), attributable to $O^6$mG, amounts to 0.065 and 0.495 AU for L and S fragments, respectively. Similarly, in the presence of random MMS lesions (mGC+MMS), incorporation, above background (dotted line in *Figure 3—figure supplement 1D*), attributable to $O^6$mG, amounts to 0.115 and 1.17 AU for L and S fragments, respectively. These results clearly show that $O^6$mGC-induced repair essentially takes place within the S fragment, with only modest spill-over into the L fragment (10–15%). This observation appears to be in good agreement with the estimated average MMR patch size (~500 nt). Thus, MMS adducts do not modify the repair pattern, that is, the relative distribution of $^{32}$P incorporation in S and L fragments, but they increase the frequency of repair centered at $O^6$mG sites. In conclusion, we show that stimulation of repair synthesis by N-alkyl adducts specifically occurs in the vicinity of the $O^6$mG adducts, illustrating that processing of N-alkyl adducts enhances MMR activity.

## MNU-treated plasmids undergo double-strand breaks during incubation in extracts

Work in *E. coli* provided elegant genetic evidence that the cytotoxicity of alkylating agents forming $O^6mG$ adducts (such as N-methyl-N'-nitrosoguanidine and MNU), including formation of replication-independent DSB, was strongly influenced by the status of the MMR pathway (*Karran and Marinus, 1982*; *Nowosielska and Marinus, 2008*). We wondered whether MNU can induce formation of DSBs independently of DNA replication. To increase the sensitivity of our assay toward DSB detection, we used a larger plasmid, pEL97 (11.3 kb), and treated it with MMS or MNU to introduce one alkylation event, on average, every 500 nt (*Figure 4—figure supplement 1*). We also treated one sample with double the concentration of MNU to achieve a twofold higher lesion density. Quantification of N-alkyl adducts by alkaline cleavage and subsequent agarose gel electrophoresis led to the expected lesion density of one N-alkyl adduct every 500 nt for MMS and MNU+, and one N-alkyl adduct every 250 nt for MNU++ (*Figure 4—figure supplement 1C*).

Alkylated and control plasmids (*Figure 4—figure supplement 2A*) were incubated in NPE for 60' in the presence of $\alpha^{32}$P-dATP, resolved by agarose gel electrophoresis, and visualized by ethidium bromide staining and $^{32}$P imaging (*Figure 4A*). Both MMS and MNU caused substantial conversion of the plasmid from the supercoiled to the open circular form, as expected during repair synthesis. Consistent with our results above, MNU induced much more repair synthesis than MMS. Strikingly, in both ethidium bromide and $^{32}$P images, a linear plasmid was detected after exposure to MNU, but not MMS. For a twofold increase in MNU exposure, the linear plasmid band increased approximately fourfold (*Figure 4B*). This quadratic dose-response strongly suggests that DSBs occur as a consequence of two independent repair events at neighboring lesions, for example a BER event at an N-alkyl adduct leading to a nick in one strand that is encountered by a gap formed by an MMR event initiated at an $O^6mG$ site in the opposite strand (*Figure 5*). To reveal SSBs, the same samples were denatured prior to native gel electrophoresis (*Figure 4—figure supplement 2B*). In the MNU++ sample, no linear ssDNA was left, all the DNA molecules running as a smear centered around the 3000 nt position (*Figure 4—figure supplement 2B*). The observed smear reveals that the double-stranded DNA running as open circular plasmid molecules in the neutral loading gel (*Figure 4A*) contain each, on average, three to four nicks per plasmid strand. The data reveal that repair of MNU-treated plasmids in NPE causes SSBs and that once the density of SSBs is high enough, DSBs result.

## Discussion

With respect to the biological responses to SN1 alkylating agents, most attention has so far been devoted to responses that occur in the first or second cell cycle following treatment as mentioned in the 'Introduction' section (*Noonan et al., 2012*; *Plant and Roberts, 1971*; *Quiros et al., 2010*).

In the present paper, we focus on early processing of DNA alkylation adducts by repair pathways before the event of replication. Interestingly, we identified the formation of DSB as the result of a putative crosstalk between repair pathways.

### Late responses to SN1 agents

Response of cells to SN1 methylating agents was shown to be initiated at $O^6mG$:T mispairs that form upon DNA replication of $O^6mG$-containing DNA template and shown to involve the MMR machinery (*Goldmacher et al., 1986*; *Day et al., 1980*; *Karran et al., 1993*; *Yarosh et al., 1983*). The $O^6mG$:T mispair is efficiently recognized by MutSα, the key MMR initiator protein. Following removal of the nascent T residue across $O^6mG$, T will be re-inserted at a high frequency during the MMR gap-filling step, thus re-forming the initial $O^6mG$:T mispair. This iterative process, called 'futile cycling', has received experimental support (*Mazon et al., 2010*; *York and Modrich, 2006*). However, it is not yet clear how these futile cycles lead to DSBs (*Ochs and Kaina, 2000*), apoptosis, and cell death (*Gupta and Heinen, 2019*; *Kaina and Christmann, 2019*). Two mutually non-exclusive models have been proposed: (i) a direct model where the encounter of the replication fork with the MMR intermediates leads to fork collapse and to subsequent cytotoxic events and (ii) a signaling model where the MutSα complex acts as a sensor leading to ATR recruitment and subsequent initiation of the ATR-Chk1 signaling pathway (*Duckett et al., 1999*; *Yoshioka et al., 2006*). However,

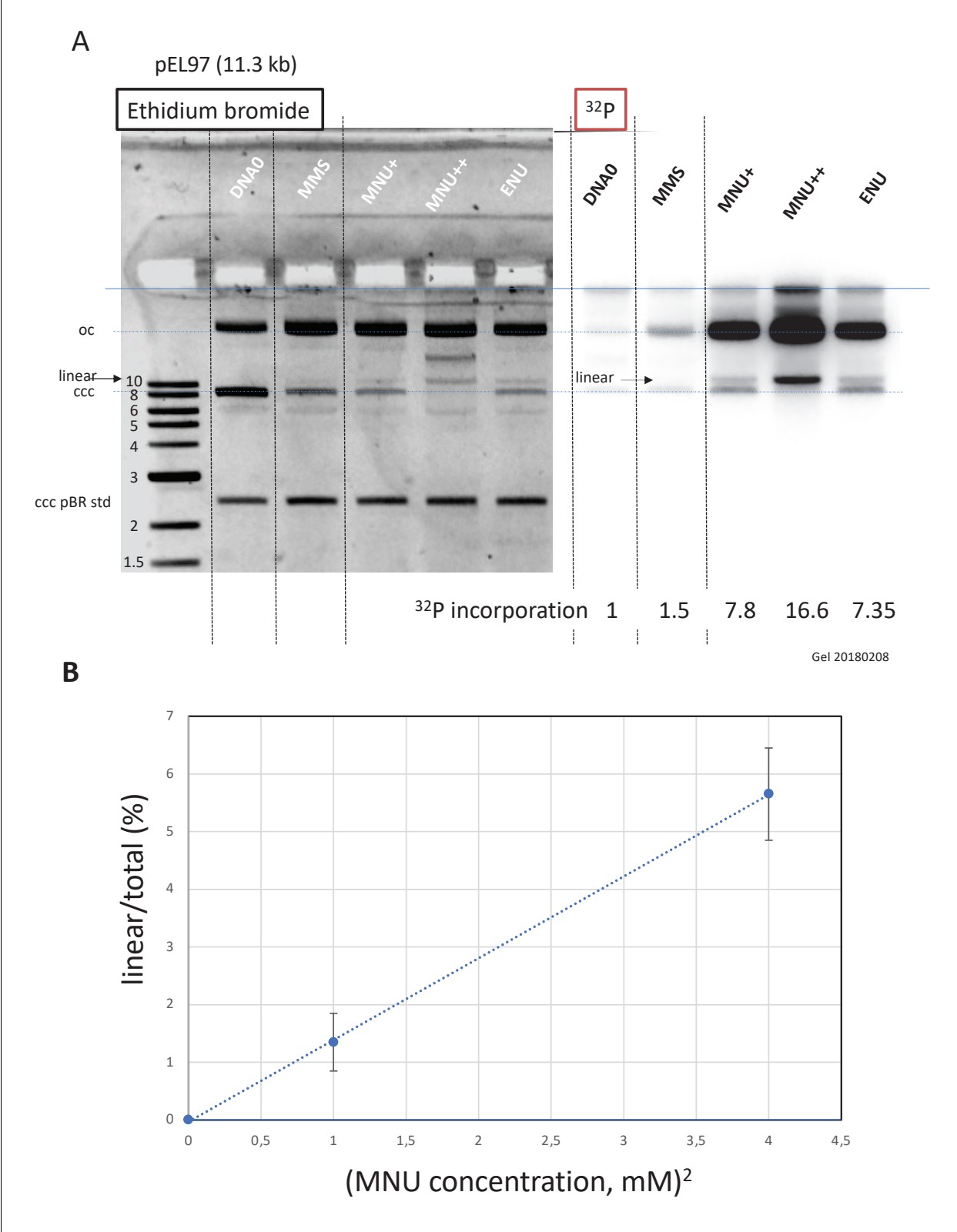

**Figure 4.** Double-strand breaks occur in MNU-treated plasmids during incubation in extracts. (**A**) Analysis by agarose gel electrophoresis (AGE) of alkylated plasmids (pEL97: 11.3 kb) incubated in nucleoplasmic extracts (NPE) in the presence of $\alpha^{32}$P-dATP. Plasmid pEL97 was treated with methyl-methane sulfonate (MMS), N-methyl-N-nitrosourea (MNU)+, and ENU as to introduce ≈ one alkylation event, on average, every 500 nt. For MNU, a plasmid with twice the level of alkylation (MNU++, one lesion every 250 nt) was also produced (**Figure 4—figure supplement 1**). Alkylation of these

*Figure 4 continued on next page*

*Figure 4 continued*

plasmids essentially not affected their migration on agarose gels (*Figure 4—figure supplement 2A*). After 2 hr of incubation, the reaction was stopped and a known amount of pBR322 (10 ng) plasmid was added as an internal standard. Ethidium bromide image: in different lanes, the internal standard band, pBR (covalently closed circular [ccc]), appears to be of similar intensity (1158 +/- 95 arbitrary units [AU]), assessing reproducible DNA extraction. For the alkylated plasmids, incubation in NPE led to massive conversion from ccc to relaxed plasmids. $^{32}$P image: little incorporation of $^{32}$P-dATP is seen in DNA0 and in MMS-treated plasmids compared to MNU- and ENU-treated plasmids as shown by the relative incorporation levels normalized to one for untreated plasmid (DNA0). As expected, the MNU++ sample exhibits about twice the amount of incorporated radioactivity compared to MNU +. In both ethidium bromide and $^{32}$P images, a small amount of linear plasmid is seen mostly in the MNU++ sample. This band is also visible in the MNU+ and ENU lanes although at a weaker intensity. (**B**) Quadratic dose-response for double-strand break (DSB) formation. When the % of linear form (linear/(linear + oc)) is plotted as a function of the square dose of MNU (mM$^2$) for untreated, MNU+, and MNU++ plasmids, we observed a straight line ($y = 1.4173x - 0.0288$; $R^2 = 0.9999$).

The online version of this article includes the following figure supplement(s) for figure 4:

**Figure supplement 1.** Estimation of N-alkylation levels of modification by MMS and MNU.

**Figure supplement 2.** Fragmentation of alkylated plasmid as analyzed on AGE loaded under alkaline conditions.

presently, there is more evidence that the critical cytotoxic response to methylating agents is the consequence of direct MMR processing rather than being mediated by a mere signaling model (*Cejka and Jiricny, 2008*; *Karran, 2001*; *Liu et al., 2010*; *York and Modrich, 2006*).

## Early responses to SN1 agents

While all biological responses described above require replication of O$^6$mG-containing DNA templates as the first step, we wanted to investigate the processing of MNU-alkylated DNA in the absence of replication. Interestingly, we detected robust, MMR-dependent, UDS upon incubation of MNU-treated plasmids in NPE. This observation reveals that, not only are O$^6$mG:C lesions recognized by MutSα as previously noted (*Duckett et al., 1999*; *Karran et al., 1993*), but also the whole MMR repair process is engaged and proceeds to completion. We would also like to stress the high sensitivity of the pull-down assay with respect to MMR protein capture. Indeed, the whole MMR machinery is enriched (*Figure 1B*) using a plasmid that on average carries only 2–3 O$^6$mG lesions/plasmid. In striking contrast, despite their abundance, ≈26 N-alkyl lesions/plasmid, N-alkyl lesions only recruit few specific proteins (*Figure 1C*).

We wanted to investigate the potential effect that N-alkyl adducts may have on MMR processing at O$^6$mG:C base pairs. For that purpose, we compared a plasmid with a single site-specific O$^6$mG:C lesion to a plasmid additionally treated with MMS, an agent known to induce essentially only N-alkyl adducts. The MMS treatment was adjusted as to produce the same amount of N-alkyl adduct as generated by MNU. The single-adducted O$^6$mG:C plasmid triggered MMR-mediated repair synthesis centered around the O$^6$mG adduct. Interestingly, the presence of randomly distributed N-alkyl adducts led to a threefold increase of the MMR repair activity in the vicinity of the O$^6$mG adduct. These data raise two questions: first, how the MMR machinery gets engaged in ccc plasmid and second, how MMR activity is further stimulated by N-alkyl adducts. In current models, functional engagement of MMR involves a mismatch recognized by MutSα, and the subsequent recruitment of MutLα and PCNA (*Jiricny, 2006*). Loading of PCNA by RFC normally requires a single-stranded nick but it was also shown to occur, although less efficiently, on ccc DNA (*Pluciennik et al., 2013*; *Pluciennik et al., 2010*). Under these conditions, PCNA loading and MMR processing lack strand directionality. With respect to the mechanism by which MMR activity, at a single O$^6$mG:C lesion, becomes stimulated several folds by the presence of N-alkyl adducts, we propose that processing of N-alkyl adducts by BER creates repair intermediates (nicks) that stimulate PCNA loading. It was previously shown that BER intermediates formed at oxidized purines or U residues can stimulate MMR processing (*Repmann et al., 2015*; *Schanz et al., 2009*).

## DSBs form in MNU-treated DNA in the absence of replication: potential therapeutic significance

Interestingly, incubation of MNU-treated plasmids in extracts leads to DSBs (*Figure 4*) that arise with a quadratic dose-response, suggesting the occurrence of two independent repair activities taking place simultaneously in opposite strands at lesions that may be up to several hundred

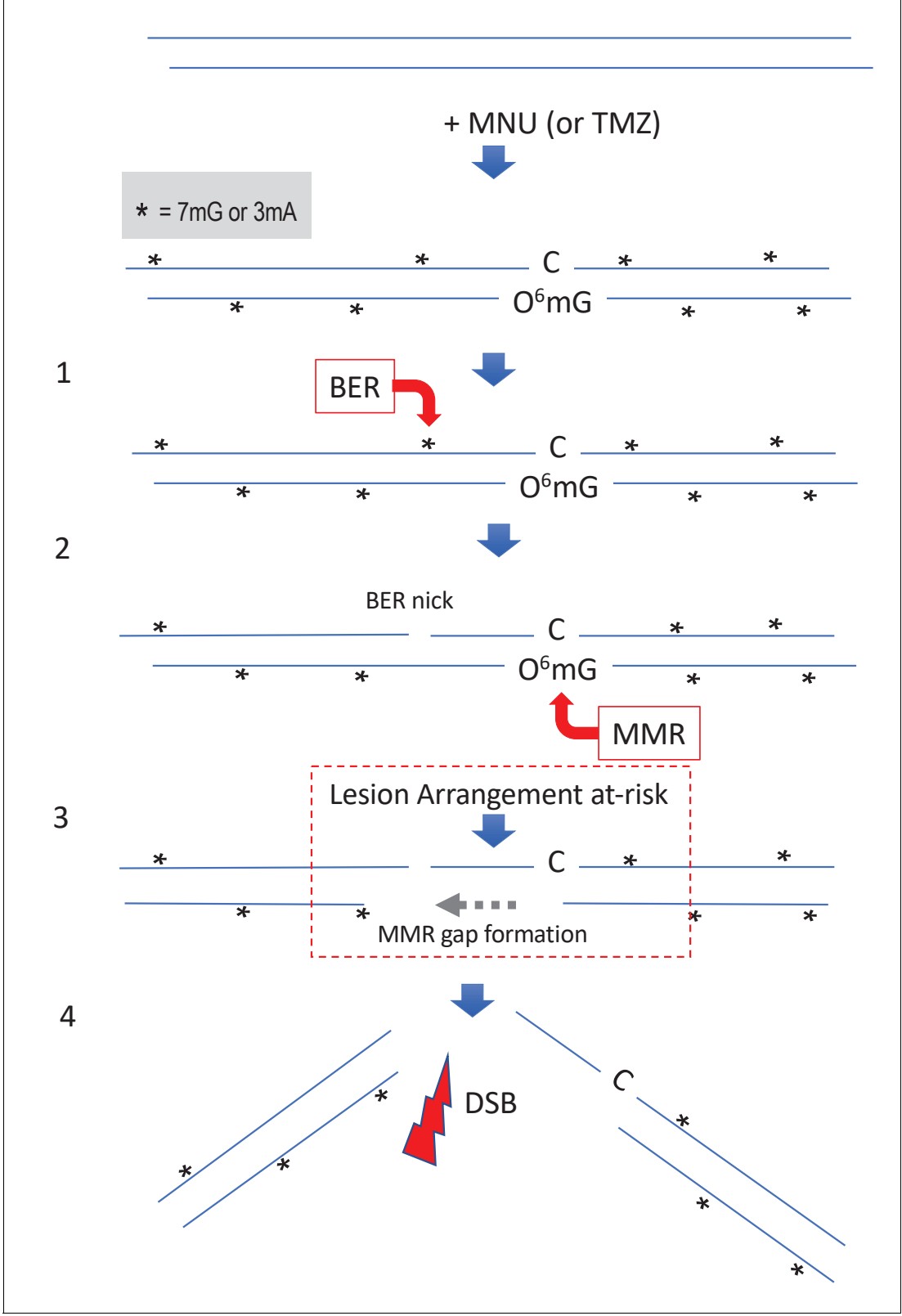

**Figure 5.** Simultaneous repair of two closely spaced MNU-induced lesions may lead to a DSB. Such a situation occurs when an N-alkyl lesion located within ≈500 nt of an $O^6mG$ lesion is processed simultaneously ('Lesion Arrangement at-risk'). Note that the mismatch repair (MMR) excision track can occur on either strand as described for noncanonical MMR (*Peña-Diaz et al., 2012*). Reaction of N-methyl-N-nitrosourea (MNU) with double-stranded DNA induces N-alkylation adducts, mostly 7mG and 3mA shown as * and O-alkylation adducts ($O^6mG$), at a ratio of 10:1 approximately. Step 1: a base

*Figure 5 continued on next page*

*Figure 5 continued*

excision repair (BER) event is initiated at an N-alkyl adduct, creating a nick. Step 2: concomitantly, an MMR event takes place, in the opposite strand, at a nearby $O^6$mG:C site. Step 3: the MMR machinery extends the nick into a several hundred nt-long gap by means of Exo1 action. Step 4: the two independently initiated repair events lead to a double-strand break (DSB), if the MMR gap reaches the BER-initiated nick before resealing.

nucleotides apart (see scenario in *Figure 5*). Similarly, in vitro processing of neighboring G/U mispairs by BER and ncMMR was shown to lead to DSBs (*Bregenhorn et al., 2016*).

The extent of DNA alkylation triggered by MNU in vitro, as deduced from our alkaline cleavage determination, fits surprisingly well with the amount of alkylation induced by TMZ in cells at equal concentrations (*Moody and Wheelhouse, 2014*). According to the model (*Figure 5*), formation of a DSB may occur when an N-alkyl lesion is located within the repair track mediated by MMR at an $O^6$mG:C site. In the clinic, a daily dose of TMZ results in 50 µM serum concentration and was shown to induce $5.2 \times 10^4$ and $7.3 \times 10^5$ O-alkyl and N-alkyl lesions per human genome, respectively (Kaina, personal communication). We can estimate the number of events (per genome) where an N-alkyl lesion is located within 500 nt on either side of an $O^6$mG:C site. Given the N-alkyl lesion density ($7.3 \times 10^5/6 \times 10^9 = 1.2 \times 10^{-4}$), the probability of presence of an N-alkyl lesion within an MMR track is 0.12. In other words, among the $5.2 \times 10^4$ $O^6$mG lesions, $\approx 6240$ are likely to contain an N-alkyl lesion within a 1000 nt excision track. We will refer to such a lesion configuration as a 'Lesion Arrangement at-risk' for DSB formation.

Let us now estimate the level of DSB that may occur in a human genome, by extrapolation of our in vitro data. In the present work, $\approx 6\%$ of plasmid DNA (11.3 kb) treated by MNU at 2 mM exhibits a DSB (*Figure 4*). The observed amount of DSBs may in fact only reflect a steady-state level since efficient re-ligation mechanisms are known to operate in NPE (*Graham et al., 2016*). As MNU and TMZ exhibit similar reactivities (*Moody and Wheelhouse, 2014*) (Kaina personal communication), a dose of 2 mM MNU would lead to $3 \times 10^9 \times 0.06/11,300 = 16,000$ DSBs per genome. In the clinic, the level of TMZ in the serum reaches up to 50 µM, that is 40 times less than the 2 mM dose used in vitro. Given the quadratic dose-response, the estimated amount of DSBs per genome would be 1600 times less, that is $\approx 10$. Let's note that the conversion rate of a lesion arrangement at-risk into an actual DSB appears to be quite low ($10/6240 \approx 0.16\%$), reflecting the requirement for simultaneous occurrence of two repair events (MMR and BER).

The alkylating agent TMZ, a chemical mimic of MNU, is presently the first-line and only anti-cancer drug in glioblastoma therapy (*Moody and Wheelhouse, 2014*). The cytotoxic mode of action of alkylating agents such as TMZ is believed to result from iterative MMR cycles. Iterative MMR cycles are deemed to lead to DSBs via a mechanism that is not yet established (*Ochs and Kaina, 2000*). Indeed, it is not known whether DSBs occur spontaneously at these sites or as a consequence of the replication fork running into the MMR intermediates. Induction of these putative DSBs is presently thought to be the primary mode-of-action of TMZ when administered to patients with glioblastoma.

Understanding both early and late cellular responses to MNU/TMZ appears thus to be critical. During cancer treatment, a dose of TMZ is delivered concomitantly with a radiotherapy session daily, for 6 weeks (for a recent review, see *Strobel et al., 2019*). As estimated above, a daily dose of TMZ may lead to an $\approx 10$ DSBs/cell resulting from BER/MMR crosstalk, a number comparable to the number of DSBs induced by 0.5–1 Gy of ionizing radiations (IR). Moreover, it was established empirically that the treatment TMZ plus radiotherapy exhibits supra-additive cytotoxicity as long as TMZ administration *precedes* radiotherapy (*Bobola et al., 2010*). Our data may provide some rationales for this empirically determined regimen. Indeed, the ssDNA stretches formed at early time points during MMR processing at $O^6$mG:C sites (step three in *Figure 5*) constitute preferential targets for the conversion of the numerous SSBs induced by IR into DSBs, thus providing an explanation for the observed supra-additivity in the treatment when TMZ *precedes* IR. A commonly used radiotherapy session involves an IR dose of 2 Gy that predominantly induces $\approx 2000$ SSBs and $\approx 40$ DSBs/cell.

As the majority of cells in a glioblastoma tumor are not proliferating, insights into attacking non-dividing cells might be very useful in treating this almost always fatal tumor. This pre-replicative mechanism for TMZ cytotoxicity will need to be investigated in cellular systems. In conclusion, the present work offers a novel mechanistic insight into the cytotoxicity of TMZ via induction of DSBs, at early time points following exposure, before replication. This early response comes in complement to the late, replication and cell cycle-dependent, responses that have been described over the years.

## Materials and methods

### Plasmids

Alkylated plasmids as used in the present paper are outlined below.

| Name | Size (kb) | Assay |
|------|-----------|-------|
| pAS200.2 | 2.1 | Site-specific $O^6$mG lesion |
| pBR322 | 4.3 | Random alkylation/UDS repair assay |
| pAS04 | 6.5 | IDAP pull-down assay/MS analysis of bound proteins |
| pEL97 | 11.3 | Random alkylation/UDS assay/post-incubation analysis |

Akylation reactions were conducted as indicated in *Figure 1—figure supplement 1*, at a plasmid concentration of 10 ng/µl in CE buffer (citrate 10mM, pH 7, ethylenediaminetetraacetic acid (EDTA) 1 mM) + 10% dimethyl sulfoxide final.

Alkylation reactions were terminated by addition of STOP buffer (5x: 1.5 M sodium acetate, 1 M mercapto-ethanol) followed by ethanol precipitation. The DNA pellet was washed with ethanol 90% and re-dissolved in TE at 50 ng/µl.

### Cleavage reactions at 7-alkylG and 3-alkylA adducts

Alkylated plasmids (50 ng in 10 µl of CE buffer) were first incubated for 90°C during 15′ at pH 7 (PCR machine). Following addition of 1 µl of NaOH 1N, the sample was further incubated at 90°C for 30′. Following addition of 2 µl of alkaline 6x loading buffer (NaOH 300 mM, EDTA 6 mM, Ficoll (Pharmacia type 400) 180 mg/ml, 0.15% (w/v) bromocresol green, 0.25% (w/v) xylene cyanol), the cleaved plasmid samples were loaded on a neutral agarose gel (*Figure 1—figure supplement 1A*).

### NPE and HSS *Xenopus* extracts

Two types of extracts derived from *Xenopus laevis* eggs were used throughout the paper, namely NPE and HSS, as described previously (*Lebofsky et al., 2009*).

### Western blot

Antibodies used against Mlh1, Pms2, and Pms1 are as previously described (*Kato et al., 2017*; *Kawasoe et al., 2016*). For western blotting, primary antibodies were used at 1:5000 dilution.

### Single adducted plasmids

Covalently closed circular plasmids containing a site-specific $O^6$mG:C base pair (plasmid mGC) and the corresponding lesion-free control (plasmid GC) were constructed. We also constructed similar plasmids with a single GT or a single $O^6$mG:T mismatch located at the same position (plasmids GT and mGT, respectively). All constructs were derived from the plasmid vector pAS200.2 (2.1 kb) (*Isogawa et al., 2020*).

### Plasmid immobilization on magnetic beads and pull-down procedure

Alkylated plasmid samples (250 ng of each -MMS, -MNU, and -ENU), as well as a non-alkylated control sample (DNA0), were immobilized on magnetic beads at a density of 10 ng plasmid/µl of M280 bead slurry using a triple helix-based capture methodology (*Isogawa et al., 2018*). The TFO1 probe used here was 5′ Psoralen – C6 – TTTTCTTTTCTCCTCTTCTC– C124 – Desthiobiotin (20 mer) with C124:hexaethylene glycol ×6. Underlined C is for 5mC; it was synthesized by using DNA/RNA automated synthesizer and purified with conventional methods (*Nagatsugi et al., 2003*).

Immobilized plasmid DNA was incubated in NPE (final volume, 16 µl) for 10 min at RT under mild agitation. To monitor non-specific protein binding to beads, we included a negative control (noDNA sample) containing the same amount of M280 beads treated under the same conditions but in absence of plasmid DNA. Reactions were stopped by addition of 320 µl of a 0.8% HCHO solution to cross-link the protein-DNA complexes for 10 min at RT. The beads were subsequently washed at RT with 200 µl of 100 mM NaCl-containing buffer (ELB buffer), re-suspended in 70 µl of extract dilution

buffer, and layered on top of a 0.5 M sucrose cushion in Beckman Coulter tubes (Ref: 342867). The beads were quickly spun through the cushion (30 s at 10,000 rpm), the bead pellet re-suspended into 40 µl of ELB sucrose, and further analyzed by PAGE or by MS.

## PAGE/silver staining

An aliquot of each incubation experiment, corresponding to 30 ng of immobilized plasmid, was treated at 99°C for 25 min in a PCR machine to revert HCHO cross-linking in LLB, 50 mM dithiothreitol (DTT). Samples were loaded on a 4–15% PAGE (Biorad pre-cast) gel at 200 volts for 32 min and stained using the silver staining kit (silver StainPlus, Biorad).

## Incorporation of $\alpha^{32}$P-dATP into DNA: spot assay

Plasmids were incubated in nuclear extracts supplemented with $\alpha^{32}$P-dATP; at various time points, an aliquot of the reaction mixture was spotted on DEAE paper (DE81). The paper was soaked in 100 ml 0.5 M $Na_2HPO_4$ (pH ≈ 9) and shaked gently for 5′ before the buffer was discarded; this procedure was repeated twice. Finally, the paper was washed for an additional two times in 50 ml ethanol, air dried, and analyzed by $^{32}$P imaging and quantification. The extent of DNA repair synthesis is expressed as a fraction of input plasmid replication (i.e., 10% means that the observed extent of repair synthesis is equivalent to 10% of input plasmid replication). This value is determined knowing the average concentration of dATP in the extracts (≈ 50 µM) and the amount of added $\alpha^{32}$P-dATP.

## Mass spectrometry

Label-free MS analysis was performed using on-bead digestion. In-solution digestion was performed on beads from plasmid pull-downs. We added 20 µl of 8 M urea, 100 mM EPPS, pH 8.5, to the beads, then 5 mM TCEP, and incubated the mixture for 15 min at RT. We then added 10 mM of iodoacetamide for 15 min at RT in the dark. We added 15 mM DTT to consume any unreacted iodoacetamide. We added 180 µl of 100 mM EPPS, pH 8.5, to reduce the urea concentration to <1 M, 1 µg of trypsin, and incubated at 37°C for 6 hr. The solution was acidified with 2% formic acid and the digested peptides were desalted via StageTip, dried via vacuum centrifugation, and reconstituted in 5% acetonitrile, 5% formic acid, for liquid chromatography (LC)-MS/MS processing.

All label-free mass spectrometry data were collected using a Q Exactive mass spectrometer (Thermo Fisher Scientific, San Jose, CA) coupled with a Famos Autosampler (LC Packings) and an Accela600 LC pump (Thermo Fisher Scientific). Peptides were separated on a 100-µm inner diameter microcapillary column packed with ~20 cm of Accucore C18 resin (2.6 µm, 150 Å; Thermo Fisher Scientific). For each analysis, we loaded ~2 µg onto the column. Peptides were separated using a 1 hr gradient of 5–29% acetonitrile in 0.125% formic acid with a flow rate of ~300 nl/min. The scan sequence began with an Orbitrap MS1spectrum with the following parameters: resolution 70,000, scan range 300–1500 Th, automatic gain control (AGC) target $1 \times 10^5$, maximum injection time 250 ms, and centroid spectrum data type. We selected the top 20 precursors for MS2 analysis, which consisted of high-energy collision dissociation (HCD) with the following parameters: resolution 17,500, AGC $1 \times 10^5$, maximum injection time 60 ms, isolation window 2 Th, normalized collision energy (NCE) 25, and centroid spectrum data type. The underfill ratio was set at 9%, which corresponds to a $1.5 \times 10^5$ intensity threshold. In addition, unassigned and singly charged species were excluded from MS2 analysis and dynamic exclusion was set to automatic.

## Mass spectrometric data analysis

Mass spectra were processed using a Sequest-based in-house software pipeline. MS spectra were converted to mzXML using a modified version of ReAdW.exe. Database searching included all entries from the *X. laevis*, which were concatenated with a reverse database composed of all protein sequences in the reverse order. Searches were performed using a 50 ppm precursor ion tolerance. Product ion tolerance was set to 0.03 Th. Carbamidomethylation of cysteine residues (+57.0215 Da) was set as static modifications, while oxidation of methionine residues (+15.9949 Da) was set as a variable modification. Peptide spectral matches (PSMs) were altered to a 1% FDR (*Elias and Gygi, 2010*; *Elias and Gygi, 2007*). PSM filtering was performed using a linear discriminant analysis, as described previously (*Huttlin et al., 2017*), while considering the following parameters: XCorr, ΔCn, missed cleavages, peptide length, charge state, and precursor mass accuracy. PSMs were identified,

quantified, and collapsed to a 1% FDR and then further collapsed to a final protein-level FDR of 1%. Furthermore, protein assembly was guided by principles of parsimony to produce the smallest set of proteins necessary to account for all observed peptides.

## Acknowledgements

We thank Johannes Walter (Harvard Medical School) for providing space, advice, and materials, and Paul Modrich (Duke Univ) and Bernd Kaina (Medical Univ, Mainz) for insightful reading and suggestions. This work was partially supported by NIH/NIGMS grant R01 GM132129 (to JAP) and MEXT/JSPS KAKENHI JP20H03186 and JP20H05392 (to TT).

## Additional information

### Funding

| Funder | Grant reference number | Author |
| --- | --- | --- |
| National Institutes of Health | R01 GM132129 | Joao A Paulo |
| Japan Society for the Promotion of Science | JP20H03186 | Tatsuro Takahashi |
| Japan Society for the Promotion of Science | JP20H05392 | Tatsuro Takahashi |

The funders had no role in study design, data collection and interpretation, or the decision to submit the work for publication.

### Author contributions

Robert P Fuchs, Conceptualization, Data curation, Formal analysis, Investigation, Methodology, Writing - original draft, Writing - review and editing; Asako Isogawa, Joao A Paulo, Investigation, Methodology; Kazumitsu Onizuka, Tatsuro Takahashi, Resources; Ravindra Amunugama, Resources, Investigation; Julien P Duxin, Resources, Methodology; Shingo Fujii, Validation, Investigation, Writing - review and editing

### Author ORCIDs

Robert P Fuchs ⬤ https://orcid.org/0000-0003-1098-4325
Shingo Fujii ⬤ https://orcid.org/0000-0002-5800-7235

### Decision letter and Author response

Decision letter https://doi.org/10.7554/eLife.69544.sa1
Author response https://doi.org/10.7554/eLife.69544.sa2

## Additional files

### Supplementary files

- Source data 1. Original image data used in Figures and Figure supplements.
- Source data 2. Image data before trimming used in Figures and Figure supplements.
- Source data 3. Resource data of MS analysis.
- Transparent reporting form

### Data availability

Source data files have been provided for MS data, gels and blots in main or figure supplements.

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
