## [Decision Letter]

**Acceptance summary:**

Glioblastomas, like many tumors, consist of a cohort of actively dividing cells and a substantially larger fraction of non-proliferating cells. The standard of care involves administration of a chemotherapy drug (temozolomide (TMZ)) whose antitumor activity is thought to be dependent on a toxic intermediate produced during DNA replication. This reports highlights the importance of the O6mG:C lesion independent of replication which leads to DSBs through coupling of MMR and BER.

**Decision letter after peer review:**

Thank you for submitting your article "Crosstalk between repair pathways elicits Double-Strand Breaks in alkylated DNA: implications for the action of temozolomide" for consideration by *eLife*. Your article has been reviewed by 3 peer reviewers, and the evaluation has been overseen by a Reviewing Editor and Kevin Struhl as the Senior Editor. The following individuals involved in review of your submission have agreed to reveal their identity: Wolf-Dietrich Heyer (Reviewer #1); Bennett Van Houten (Reviewer #2); Michael M Seidman (Reviewer #3).

This paper calls attention to an overlooked feature of TMZ chemotherapy- that intermediates produced by two separate DNA repair pathways, operating on different adducts produced by the same compound, can produce a potentially toxic structure (DSBs). The message of the paper would be strengthened, and have greater impact, by a few additional experiments that would further support the contribution of BER in the extract system and provide some insight as to how to exploit these findings.

However, the reviewers consider the claims to be sufficiently supported by the present data. No new experiments are required for the resubmission, although the authors may want to consider enhancing the impact of their study.

Essential revisions:

1) The authors state on page 6 without showing the data that the MGMT inhibitor Patrin-2 has no effect on DNA repair-associated DNA synthesis in their system. This contradicts earlier findings in *Xenopus* egg extracts by Olivera Harris et al. 2015 DNA Repair. This discrepancy cannot be left unexplained, as direct reversal affects the interpretation of the data in this manuscript.

2) A table of specific plasmids, lengths and the assays/justification that they were used for would be helpful to the reader as the authors switch from one plasmid to another in different experiments and it is not always clear which plasmid is being used in which experiments.

3) Figure 2B, page 6: What is evidence that only less than 30% of O6mG sites are substrates for MMR?

4) The data in supplemental Figure 2 is intriguing. However, I am having difficult reconciling the repair synthesis shown in Figure 2 supplement 1 with supplement 2. Why was NPE used in one experiment and HSS in the other? What happens when human AAG is added back?

5) Page 6 bottom, Page 7 top: What is the rationale for using different extracts? It is not always clear why and when the used total *Xenopus* extracts versus the supernatant from a high-speed centrifugation (HSS). Is it possible that some DNA binding proteins are spun down in the pellet?

6) Figure 1 should have an accompanying series of tables showing ALL the data from these volcano plots with proper annotation and p-values for significance. Just showing dots on a plot is not adequate. Also was the same experiment repeated with both nuclear extracts and the HSS? A direct comparison would be helpful.

7) Figure 4A: What is the evidence that the bands labeled linear are indeed linear? The interpretation rests on the migration behavior but additional evidence is needed to show these species are linear. For example, are they sensitive to ExoV or other dsDNA exonucleases.

8) Figure 4B: A major conclusion rests on the quadratic relationship between linear DNA and MNU concentration, but the titration has only two data points. Is this sufficient?

9) The figures measuring repair synthesis show ethidium stained gels that display the covalently closed circular and open circular forms of the plasmids. In light of the substantial labeling of control (unmodified) plasmids it would be helpful to show the plasmids after exposure to MMS or MNU and before they are introduced to the extracts. This was done in Figure 4 supplement 2A in which open circular forms are clearly present in the plasmid samples. Given the importance of nicks for MMR, and as potential targets for opportunistic polymerases, pre-existing nicks could be a confounding variable in the interpretation of "repair" synthesis.

[Editors' note: further revisions were suggested prior to acceptance, as described below.]

Thank you for resubmitting your work entitled "Crosstalk between repair pathways elicits Double Strand Breaks in alkylated DNA and implications for the action of temozolomide" for further consideration by *eLife*. Your revised article has been evaluated by Kevin Struhl (Senior Editor) and a Reviewing Editor.

The manuscript has been improved but there are some remaining issues that need to be addressed, as outlined below:

In their rebuttal and in this revision, the authors clarified many of the concerns raised and made a number of text changes. However, the revision would have benefited from more extensively addressing these issues.

1) Discrepancy with earlier work: It would be helpful to the reader to incorporate the rebuttal response about the Hsieh et al. paper in the manuscript.

2) Essential revision #1: This clarification is fine, but this should be stated in the manuscript, in particular the argument about the underestimate of the UDS.

3) Essential revision #3: The clarification is helpful, and the GT/mGT data should be added to the manuscript.

4) Point #12: The addition to the discussion is minimal and should be expanded.

---

## [Author Response]

This paper calls attention to an overlooked feature of TMZ chemotherapy- that intermediates produced by two separate DNA repair pathways, operating on different adducts produced by the same compound, can produce a potentially toxic structure (DSBs). The message of the paper would be strengthened, and have greater impact, by a few additional experiments that would further support the contribution of BER in the extract system and provide some insight as to how to exploit these findings.However, the reviewers consider the claims to be sufficiently supported by the present data. No new experiments are required for the resubmission, although the authors may want to consider enhancing the impact of their study.Essential revisions1) The authors state on page 6 without showing the data that the MGMT inhibitor Patrin-2 has no effect on DNA repair-associated DNA synthesis in their system. This contradicts earlier findings in *Xenopus* egg extracts by Olivera Harris et al. 2015 DNA Repair. This discrepancy cannot be left unexplained, as direct reversal affects the interpretation of the data in this manuscript.

MGMT inhibition issue: as mentioned in the text, upon addition of Patrin to the NPE extract, we did not see any increase in UDS in MNU treated plasmid. There are two formal possibilities: (i) the number of MGMT molecules present in the extract is small compared to the number of O^6^mG lesions introduced in the incubation mix or (ii) our Patrin inhibitor sample is inactive. In all cases, if partial demethylation of O^6^mG by MGMT occurs, the observed amount of UDS would be under-estimated. Thus, the conclusion reached in the paper, O^6^mG:C sites are substrates for MMR, remains correct.

2) A table of specific plasmids, lengths and the assays/justification that they were used for would be helpful to the reader as the authors switch from one plasmid to another in different experiments and it is not always clear which plasmid is being used in which experiments.

At the beginning of Materials and methods, we have introduced a table listing the different plasmids and their respective usage. Moreover, the name of the plasmid that was used is now indicated in each figure.

3) Figure 2B, page 6: What is evidence that only less than 30% of O6mG sites are substrates for MMR?

The hypothesis that ≈30% of 06mG are MMR substrates is based on the data described below. We have monitored UDS activity in plasmids containing a single GT or mGT mispair. These plasmids have been treated (or not) with MMS similarly to the description of the corresponding GC or mGC constructs. The GT, mGT, GT+MMS and mGT+MMS data have been added to the corresponding GC data shown in Figure 3B.

The main observation is that GT containing constructs trigger a much stronger MMR response than their GC counterparts. In the absence of MMS, at the 120’ time point, the level of UDS in mGC represents 23% of the level in mGT. In the presence of MMS, at the 120’ time point, the level of UDS in mGC+MMS represents 38% of the level in mGT+MMS. If one supposes that mGT mispair are fully repaired by MMR (close to 100%), then repair at mGC mispairs is in the range of 1 in 3 (30-40%).

The main emphasis of the paper is to describe the existence of MMR at mGC sites (in the absence of replication). As GT and mGT mispairs only occur following replication we decided not to include these data in the initial version of the paper. We are open to discuss whether the GT ad mGT data should or not be added.

4) The data in supplemental Figure 2 is intriguing. However, I am having difficult reconciling the repair synthesis shown in Figure 2 supplement 1 with supplement 2. Why was NPE used in one experiment and HSS in the other? What happens when human AAG is added back?

Two type of extracts derived from *Xenopus laevis* eggs are used throughout the paper, namely NucleoPlasmic Extract (NPE) and High-Speed Supernatant (HSS) as described previously (Lebofsky et al., 2009). For sake of clarity, we will avoid the terminology “egg extract” and specify throughout the text whether we used NPE or HSS. All experiments use NPE except for data in Figure 2—figure supplement 2 that use HSS. This point is clarified in the text. We have also indicated in the figures whether NPE or HSS are used.

5) Page 6 bottom, Page 7 top: What is the rationale for using different extracts? It is not always clear why and when the used total *Xenopus* extracts versus the supernatant from a high-speed centrifugation (HSS). Is it possible that some DNA binding proteins are spun down in the pellet?

We observed that, in contrast to NPE, HSS do not support UDS on MNU-treated plasmid. However, a previous paper (Harris, 2015) had shown that HSS was proficient for O6mC:C repair provided a preformed nick was present in the substrate. We reasoned that perhaps our HSS extract was deficient in AAG glycosilase and thus implemented experiments in which purified hAAG was added in order to create the BER intermediate able to stimulate MMR at the O6mG:C sites.

6) Figure 1 should have an accompanying series of tables showing ALL the data from these volcano plots with proper annotation and p-values for significance. Just showing dots on a plot is not adequate. Also was the same experiment repeated with both nuclear extracts and the HSS? A direct comparison would be helpful.

An excel table showing the details for the construction of the Volcano plots has been added. The x-axis is log base 2 of the ratio between average spectral counts for MNU divided by average spectral counts for MMS. The p-value was obtained using the TEST.STUDENT function in excel; parameters uni/bilateral and type were set to 2 and 3, respectively. The y-axis is -log base 10 of the p-values. The volcano plot shown in the xlsx file corresponds to comparison between MMS and MNU (i.e. the results shown in Figure 1—figure supplement 1C). The data have been obtained using Xenbase.

We did not perform an MS analysis comparing HSS to NPE.

7) Figure 4A: What is the evidence that the bands labeled linear are indeed linear? The interpretation rests on the migration behavior but additional evidence is needed to show these species are linear. For example, are they sensitive to ExoV or other dsDNA exonucleases.

Unfortunately, we did not perform additional experiments such as sensitivity to dsDNA exonucleases. Besides the band migrating at the position expected for the linear form of the plasmid (Figure 4A), another evidence stems from Figure 4—figure supplement 2B. Indeed, in the MNU++ lane, the ss linear forms disappear in both Ethidium bromide and ^32^P images, suggesting that the oc band in the MNU++ lane (^32^P image in Figure 4B) is formed of circular DNA having multiple nicks (or gaps). Thus, DNA carrying multiple nicks or gaps maintains a circular form under neutral condition unless closely spaced ss breaks lead to linearization. Therefore, the band indicated as a linear in Figure 4A could not maintain its circular form due to two closely spaced ss breaks that led to its linearization.

Also see response to point 8 below.

8) Figure 4B: A major conclusion rests on the quadratic relationship between linear DNA and MNU concentration, but the titration has only two data points. Is this sufficient?

We agree that additional data points would be of value. However, the 0,0 point is also an experimental point as there is no band visible in non-treated plasmid sample. Moreover, in a different experiment using pBR322 (4,3kb) instead pEL97 (11,3kb) we observe a band migrating at 4,3kb in the MNU treated sample, thus confirming that MNU induces DSB’s.

9) The figures measuring repair synthesis show ethidium stained gels that display the covalently closed circular and open circular forms of the plasmids. In light of the substantial labeling of control (unmodified) plasmids it would be helpful to show the plasmids after exposure to MMS or MNU and before they are introduced to the extracts. This was done in Figure 4 supplement 2A in which open circular forms are clearly present in the plasmid samples. Given the importance of nicks for MMR, and as potential targets for opportunistic polymerases, pre-existing nicks could be a confounding variable in the interpretation of "repair" synthesis.

Native and alkylated plasmids before incubation in extracts exhibit a small amount of relaxed form that is similar for DNA0, MMS, MNU+ and MNU++ (Figure 4 supplement 2A). Thus, the alkylation reaction *per se* did not induce relaxation in addition to the pre-existing amount in DNA0 (except for ENU treatment). Total incorporation into DNA0 (1) is small compared to MNU+ (7.8) and MNU++ (16.6) as indicated in Figure 4A (relative amounts of incorporation). Moreover, data of Figure 4—figure supplement 2 indicate that nicks pre-existing in MNU treated plasmid are not able to stimulate MMR; only addition of hAAG triggers MMR.

[Editors' note: further revisions were suggested prior to acceptance, as described below.]

The manuscript has been improved but there are some remaining issues that need to be addressed, as outlined below:In their rebuttal and in this revision, the authors clarified many of the concerns raised and made a number of text changes. However, the revision would have benefited from more extensively addressing these issues.1) Discrepancy with earlier work: It would be helpful to the reader to incorporate the rebuttal response about the Hsieh et al. paper in the manuscript.

This issue is discussed p5.

2) Essential revision #1: This clarification is fine, but this should be stated in the manuscript, in particular the argument about the underestimate of the UDS.

This issue is discussed p6.

3) Essential revision #3: The clarification is helpful, and the GT/mGT data should be added to the manuscript.

GT and mGT data have been introduced in Figure 3B; GT and mGT plasmid construction has been introduced in Mat & Methods; GT and mGT UDS data are discussed p 8

4) Point #12: The addition to the discussion is minimal and should be expanded.

Point 12 is raised in the Discussion section p 11. As the mechanism of DSB formation as a result of futile cycling has not been established precisely we did not want to discuss it at length.